# Antibacterial and Antibiofilm Effect of Unifloral Honeys against Bacteria Isolated from Chronic Wound Infections

**DOI:** 10.3390/microorganisms11020509

**Published:** 2023-02-17

**Authors:** Viktória L. Balázs, Lilla Nagy-Radványi, Erika Bencsik-Kerekes, Regina Koloh, Dina Szabó, Béla Kocsis, Marianna Kocsis, Ágnes Farkas

**Affiliations:** 1Department of Pharmacognosy, Faculty of Pharmacy, University of Pécs, 7624 Pécs, Hungary; 2Department of Microbiology, Faculty of Science and Informatics, University of Szeged, 6726 Szeged, Hungary; 3Department of Medical Microbiology and Immunology, Medical School, University of Pécs, 7624 Pécs, Hungary; 4Department of Plant Biology, Institute of Biology, University of Pécs, 7624 Pécs, Hungary

**Keywords:** acacia honey, chestnut honey, goldenrod honey, linden honey, milkweed honey, antibacterial effect, antibiofilm activity, wound healing, quorum sensing

## Abstract

Honey is known as an alternative remedy for the treatment of wounds. To evaluate the potential of five Hungarian honey types against wound-associated bacteria, in vitro microbiological assays were conducted on *Pseudomonas aeruginosa, Staphylococcus epidermidis* and methicillin-resistant *Staphylococcus aureus* (MRSA). Minimum inhibitory concentration (MIC) was determined with the broth macrodilution method, and biofilm degradation capacity was tested with a crystal violet assay. To understand the underlying mechanisms, the effects of honey treatments were assessed on bacterial membrane integrity and quorum sensing (QS). The highest antibacterial activity, indicated by the lowest MIC values, as well as the highest biofilm inhibition rates and membrane disruption, was displayed by chestnut and linden honeys. The most sensitive bacterium was *S. epidermidis*. Bacterial membrane degradation took place 40 min after treatment with honey solutions of at least a 40% concentration. Each honey sample exhibited anti-QS activity, which was most pronounced in the case of chestnut honey. It was concluded that the antibacterial, biofilm-inhibiting and anti-QS activities of linden and chestnut honeys were superior to those of acacia, goldenrod and milkweed honeys. In addition to the floral source, the antibacterial effect of honey is influenced by the microbial species treated. The use of honey in wound treatment can be justified by its diverse antibacterial mechanisms.

## 1. Introduction

The most common types of chronic or non-healing wounds include pressure ulcers, venous ulcers and diabetic ulcers [1,2]. Bacterial colonization of the wound is an important step in the pathogenesis of chronic wounds, particularly if the bacteria involved are able to form biofilms. A microbial biofilm is a structured community of microbial cells surrounded by a self-produced polymer matrix. Biofilms can be created by bacteria, viruses or fungi, either as monomeric or polymeric cultures. Within biofilms, bacterial cells are located close to each other, which in itself promotes survival, as the process of horizontal gene transfer can take place quickly and easily. Bacterial cells are able to transfer resistance genes with the help of plasmids, thereby increasing antibiotic resistance, which complicates treatment [3,4]. Van der Waals forces and hydrogen bridges enable the strong adhesion of biofilms to biotic or abiotic surfaces with the help of glycocalyx. The glycocalyx, with enzymes and proteins produced, prevents antibacterial agents from entering the biofilm between the bacterial cells, thus enabling resistance to treatment with antibiotics and disinfectants [5,6,7].

In recent years, it has been proven that bacteria can communicate with each other, which leads to changes in gene expression. This is called the quorum sensing (QS) process, which controls the various physiological activities of bacterial cells. Quorum is the minimum cell density necessary for the formation of the stimulus-response system, and sensing refers to the detection of signal molecules that regulate the process [8]. Both Gram-negative and Gram-positive bacteria have QS mechanisms, but the signal molecules they transmit are different. Microbial cell-to-cell communication often occurs by releasing and receiving quorum-sensing molecules (QSMs). QSMs are abundant and widely distributed in natural or artificial microbial communities, of which N-acyl homoserine lactones (AHLs), as typical representatives of QSMs, could strongly affect the physiological metabolism of microorganisms. As a result of QS, it increases the expression of autoinducer synthase, so more AHL is produced, which amplifies the QS effect [9,10]. The bacteria influence, modify and control the functioning of the bacterial population arranged in the biofilm with the help of the released signal molecules. When the concentration of signal molecules reaches a threshold value that corresponds to the density of bacterial cells, specific gene expression is initiated. Quorum sensing systems only work if there is a sufficient number of bacterial cells in a given area, i.e., it is density dependent. Through the mechanism, it becomes possible to increase the pathogenicity of infections, the development of antibiotic resistance, the initiation of inflammatory processes in the body and the continuous increase of the biofilm [11,12]. As a result, the development of antibiotic resistance is accelerated [13,14].

Biofilm-related diseases develop slowly but are usually long lasting; the immune system is rarely able to suppress them. In addition, bacterial biofilms show little response to antibiotic treatments [15]. Biofilm-forming bacteria cause chronic infections with persistent tissue damage [11,16]. According to estimates, 65–70% of bacterial infections are associated with biofilm formation [11,17]. The biofilms of *Pseudomonas aeruginosa*, *Staphylococcus epidermidis* and methicillin-resistant *Staphylococcus aureus* (MRSA) are common on the surface of chronic wounds [18]. From a clinical point of view, one of the most dangerous multi-resistant nosocomial bacteria is *Pseudomonas aeruginosa* [19], which can be held responsible for at least 10% of hospital infections [20]. It is a Gram-negative opportunistic pathogen that occurs frequently both in our environment and in the human body as part of the normal bacterial flora (skin, upper respiratory tract, gastrointestinal tract) [20]. Its biofilm-forming properties play a significant role in the colonization of the wound surface and its biofilm is extremely resistant to antibiotics and disinfectants due to the production of alginate [21,22]. *S. epidermidis* is able to adapt to different microenvironments of the skin, so it can be found on the entire skin surface [23]. It is a Gram-positive, coagulase-negative (CoNS), opportunistic pathogen that can be easily isolated from areas of the body containing sebum and sweat glands or from the mucous membrane around body openings [20]. It is a common cause of nosocomial bacteremia and sepsis, and due to its mucus production, it adheres to skin and plastic surfaces with extreme affinity. The treatment of suture, implant and intravenous catheter infections is complicated by the bacteria’s multidrug resistance and biofilm-forming ability [24,25]. MRSA occurs both in the epithelium of the human outer covering and in the upper respiratory tract and is one of the most common pathogens of nosocomial infections. MRSA has several virulence factors, as a result of which it causes very diverse pathologies (skin and soft tissue infections, lung and endocarditis, food poisoning). The main problem in the treatment of infections is the strain’s resistance to most penicillin and cephalosporin derivatives. The use of glycopeptides (vancomycin, teicoplanin), linezolid, tigecycline and daptomycin, which are emerging as therapeutic alternatives, is greatly limited by the fact that strains with reduced sensitivity or resistance to these are increasingly common nowadays [26,27].

According to the above, it is absolutely important to investigate alternative remedies that offer an opportunity to reduce chronic wound infections. The wound healing activity of various uni- and multifloral honeys, associated with their antibacterial and anti-biofilm effects, was proven in several experiments. A great advantage of the therapeutic use of individual honeys is their diverse composition of active ingredients, which is largely determined by their botanical and geographical origins. The physicochemical properties of honey, such as pH, free acidity, electrical conductivity, moisture content, sugar spectrum, diastase and glucose oxidase activity, can be influenced even by the bee species that produce the honey [28]. Unlike other natural agents, the antibacterial effect of honey is not due to a specific compound or active ingredient but to their combination, so different strains of bacteria develop resistance to it to a lesser extent. The slightly acidic pH, hydrogen peroxide release and high polyphenol content also help tissue regeneration and contribute to antibacterial activity [29]. In general, the higher the hydrogen peroxide and total polyphenol content of honeys, the more they inhibit the growth of bacteria [30]. The high sugar content and high viscosity of honey greatly contribute to the inhibition of microbial growth and biofilm-forming ability [31], performing a barrier function in the infection process, thus preventing various pathogens from entering the wound. In addition, honey contributes to absorbing wound exudate, initiating the exfoliation of wounds, and keeping the wound area moist [32]. Furthermore, it has been documented that honey can inhibit the QS system of bacterial communities [33,34,35]. Recent experiments have indicated that honey could become an effective alternative, even in the elimination of antibiotic-resistant bacteria, including wound-infecting bacteria [36,37].

From the assortment of Central European honeys, there is evidence supporting the antimicrobial activity of acacia, chestnut, linden, rapeseed and sunflower honeys, but such data are scarce regarding goldenrod and milkweed honeys. Our aim was to test and compare the antibacterial and antibiofilm effects of five Hungarian honey varieties (black locust/acacia, chestnut, goldenrod, linden, milkweed) against the wound surface colonizing bacteria *P. aeruginosa*, *S. epidermidis* and MRSA. To understand the underlying mechanisms, experiments were designed to assess the effect of honey treatments on bacterial membrane integrity and QS.

## 2. Materials and Methods

### 2.1. Origin of Honey Samples

The honey samples were purchased from three local apiaries in Hungary in the year 2021. Milkweed (*Asclepias syriaca*) honeys originated from the Southern Great Plain area, while black locust/acacia (*Robinia pseudo-acacia*), linden (*Tilia* spp.), goldenrod (*Solidago gigantea*) and chestnut (*Castanea sativa*) honeys were harvested in the Southwest Transdanubium area. They were stored at room temperature (20–21 °C) in the dark for a maximum of three weeks.

### 2.2. Melissopalynological Analysis and Color Determination

The botanical origin of the honey samples was confirmed by microscopic pollen analysis following the modified method of Von der Ohe [38]. Ten grams of honey were dissolved in 20 mL of distilled water, and the mixture was vortexed with Combispin FVL-2400N (Biocenter Kft., Szeged, Hungary). The samples were centrifuged using a Neofuge 15R centrifuge device (Lab-Ex Ltd., Budapest, Hungary) at a speed of 8753× *g* for 10 min. Afterwards, the supernatant was decanted, and 10 mL of distilled water was added to the sediment. Another centrifugation phase followed (8753× *g* for 5 min). Distilled water (0.25 mL) was added to the precipitate and vortexed. Twenty milliliters of the resulting pollen suspension was pipetted onto the microscope slides. The slides were preheated to 40 °C using a heating plate (OTS 40, Tiba Kft., Győr, Hungary). The pollen preparations were placed in Kaiser’s glycerin jelly with fuchsine (Merck Life Science Ltd., Budapest, Hungary), and then the pollen grains were examined with a Nikon Eclipse E200 light microscope equipped with a Michrome 20MP CMOS digital camera (Auro-Science Consulting Kft., Budapest, Hungary). Micrographs were taken with the Capture 1.2 program at 400× magnification. We counted at least 500 pollen grains from each honey sample. The botanical source was identified at the level of plant species, genera or families. The relative frequency of pollen types was calculated as a percentage of the total number of pollen grains.

The color intensity of honey samples was determined according to the protocol of [39]. We prepared 50% *w*/*v* honey solutions (water temperature: 45–50 °C). After preparing the solutions, they were treated with ultrasound for 5 min and then filtered (0.45 µm pore size, Agilent Technologies, Milan, Italy). Absorbance was measured at 450 and 720 nm using a Shimadzu UV-1800 spectrophotometer (Shimadzu Schweiz GmbH, Reinach, Switzerland). Color intensity was calculated as the difference between absorbance at 450 and 720 nm, and the results were expressed in milliabsorbance units (mAU).

### 2.3. Cultivation of Test Bacteria

The antibacterial effect of honey samples was determined on *Pseudomonas aeruginosa* ATCC 27853, *Staphylococcus epidermidis* ATCC 12228 and methicillin-resistant *Staphylococcus aureus* ATCC 700698. Test bacteria were grown in 100 mL sterile BHI (Brain Heart Infusion, Sigma Aldrich Ltd., Budapest, Hungary), except for the MIC determination, in which case Mueller-Hinton Broth (MHB, Oxoid Ltd., London, UK) culture medium was used. Each bacterium was incubated in a shaker incubator (C25 Incubator Shaker, New Brunswick Scientific, Edison, NJ, USA) at 37 °C and at a speed of 60 rpm for 12 h [40]. The bacterial suspensions were diluted with clear BHI to the appropriate concentrations for each assay. For the anti-QS tests, the *Chromobacterium violaceum* 85WT (SZMC 6269) bacterial strain was used. The bacterium was cultivated on LB agar (Luria Bertani Broth, 10 g tryptone, 10 g NaCl, 6.6 g yeast extract, 15 g agar in 1000 mL distilled water, Sigma Aldrich Ltd., Budapest, Hungary) at 30 °C for 48 h.

### 2.4. Determination of Minimum Inhibitory Concentrations (MIC)

To investigate the minimum inhibitory concentrations (MIC), a broth macrodilution test (BDT) was used, which is commonly used in microbiological laboratories according to CLSI guidelines (Clinical & Laboratory Standards Institute). From each honey sample, a serial twofold dilution (using Mueller–Hinton Broth) was prepared from 50 to 1.56% and 40 to 1.25%. As a control of bacterial growth, honey was not added to the tubes. For the dilution series of antibiotics, a detergent was not used. Ten microliters of an overnight bacterial culture (~10^5^ cells/mL) was added to each tube and incubated at 37 °C for 24 h. Then, the tubes were plated out on 5% sheep blood agar and incubated again for 48 h. The number of bacterial colonies was compared to the controls and then the values of the minimum inhibitory concentrations (MIC) were determined. The MIC value was the concentration that could reduce the visible growth of bacteria in comparison with the controls. All tests were carried out in triplicate and under aerobic conditions.

### 2.5. Antibiofilm Capacity and Cell Viability Test

In order to prove the anti-biofilm effect of the honey sample, a crystal violet (CV) assay was used [41]. Bacterial biofilms were formed on 96-cell polystyrene microtiter plates. Two hundred microliters of 10^8^ CFU/mL cell suspensions were measured in the wells. The microtiter plate was incubated at 37 °C for 4 h, which time interval is sufficient for bacteria to adhere to the surface. After the incubation time, the non-adherent cells were washed with physiological saline solution. In the next step, the adherent cells were treated with MIC/2 concentrations of honey samples. After incubation (24 h, 37 °C) and washing with physiological saline, 200 μL of methanol was measured to the wells to fix the adherent cells. The incubation period was 15 min at room temperature (RT). After that 200 μL of 0.1% crystal violet solution was measured into the cells to dye the bacterial biofilms (20 min, RT). After 20 min, the dyed biofilm was dissolved with a 33% acetic acid solution and absorbance was measured at 595 nm with a plate reader (BMG Labtech SPECTROstar Nano, Budapest, Hungary). Crystal violet binds to negatively charged surface molecules and polysaccharides within the extracellular matrix of biofilms, thus allowing measurement of the total biomass of the biofilm in the cells of the microtiter plate. The effect of inhibiting biofilm formation is expressed using the following relation: Inhibitory rate = (1 −S/C) × 100% (C and S were defined as the average absorbance of control and sample groups, respectively) [42]. The measurements were carried out six times. In order to observe the viable bacterial cells in the biofilm unit, we performed cell viability tests. An MTT [3-(4,5-dimethylthiazol-2-yl)-2,5-diphenyltetrazolium bromide] assay was used based on He et al. [43]. The biofilms were prepared in 96-well microtiter plates. After incubation (4 h, 37 °C) the non-adherent cells were washed with physiological saline solution and the honey samples were added to the biofilms. After 24 h (37 °C) the non-adherent cells were washed and MTT (5 mg/mL) was added to the biofilms. MTT can color only viable cells. After 3 h, the lysing solution (1 N NaOH) was added to dissolve the biofilm for 2 h at room temperature [44]. The absorbance was measured at 590 nm using a plate reader. The results are expressed as percentages compared to the untreated control.

### 2.6. Scanning Electron Microscopy (SEM)

Scanning electron microscopy was used to investigate the structural modifications of biofilms after treatment with chestnut honey, which was the most effective sample in previous assays. Both Gram-positive (*S. epidermidis*) and Gram-negative (*P. aeruginosa*) bacteria were included in this investigation.

For biofilm formation, 5 mL of bacterial culture (10^8^ CFU/mL) was added to a sterilized bottle. During the preparation of the scanning electron microscopic examinations, biofilms were formed on degreased, sterilized coverslips. The coverslips were incubated in the bacterial suspensions for 4 h (37 °C). Adhesion occurred during the incubation time. After incubation time, the plates were washed with physiological salt solution, and then the chestnut honey sample was used as an inhibitor at a concentration of MIC/2 (5 mL). The control coverslips were the untreated ones. After 24 h (37 °C), the solutions were removed, and the non-adherent cells were removed with physiological saline. Then, the samples were prepared according to the SEM protocol: In order to fix the biofilm, the samples were incubated in 2.5% glutaraldehyde at RT for 2 h. In order to dehydrate the biofilms, absolute ethanol was used with 50%, 70%, 80%, 90% solutions and for 2 × 15 min (RT). The coverslips were then placed in a 1:2, 1:1, 2:1 mixture of t-butyl alcohol: absolute ethanol and then in absolute t-butyl alcohol for 1–1 h (RT). Finally, the samples were freeze-dried in t-butyl alcohol overnight. The sample was coated with a gold membrane and observed with a JEOL JSM IT500-HR scanning electron microscope (Jeol Ltd., Tokio, Japan) [45].

### 2.7. Membrane Degradation Study

The release of cellular material was examined in each bacterium. The bacterial suspension (10^8^ CFU/mL) was made in PBS (phosphate buffer saline) and its absorbance was measured at 260 nm. The bacterial cells were treated with different concentrations of honey samples (20, 40, 60%) for 1 h. As a positive control, 90% solution of honey samples was used.

In order to examine the time dependence of membrane degradation, the bacterial cells were suspended in PBS, which contained 60% honey. The treatment was made for different periods of time: 0, 10, 20, 40, 60 and 90 min. The bacterial cells were centrifuged after each treatment (Neofuge 15R, Lab-Ex Ltd., Budapest, Hungary) at 12,000× *g* for 2 min, and the absorbance of the supernatant was determined at 260 nm with a Metertech SP-8001 (Abl&e-Jasco Ltd., Budapest, Hungary) spectrophotometer. During the test, the nucleic acid in the supernatant was measured. The results were expressed as percentage values, which were compared to the untreated cells [42].

### 2.8. Anti-Quorum Sensing Effect

The synthesis of violacein in *Chromobacterim violaceum* is under QS regulation, which makes this bacterium suitable for screening compounds with an anti-QS capacity. The anti-QS activity of the honey samples was assessed through the inhibition of violacein synthesis in this model organism [45]. In this assay, the violacein pigment produced in liquid culture was extracted and detected spectrophotometrically according to the modified method of Choo, Rukayadi and Hwang (2016) [46] and Zambrano et al. (2019) [47]. Erlenmeyer flasks containing 10 mL of LB medium and honey in different concentrations (75%, 50%, 25%, 10%, 5%, 2%, 1%, 0.5%) were inoculated with 100 μL 24-h old bacterial culture with 10^8^ CFU/mL concentration. Flasks consisting of LB medium and bacterial suspension were used as positive controls, and flasks containing LB medium and honey were used as negative controls. After the preparation, samples were incubated at 30 °C for 24 h under continuous shaking at 150 rpm. Before pigment extraction, the cell number of all samples was determined by plating them on LB agar media. Bacterial colonies were counted, and the results were given in log CFU/mL. Two mL aliquots were placed in sterile Eppendorf tubes and subjected to centrifugation procedure (16,200× *g* for 20 min) in order to precipitate the insoluble violacein pigment. After discarding the supernatant, the pellet was solubilized in 1 mL of dimethyl sulfoxide (vortex for 20 min). Cellular debris was removed by further centrifugation (16,200× *g*, 20 min) and 200 μL of supernatant was added into a sterile 96-well microtiter plate. Absorbance was measured at 585 nm using a SPECTROstar Nano microplate reader (BMG Labtech, Germany). We compared the absorbance of the amount of pigment extracted from the liquid cultures of *C. violaceum* 85 WT with the measured cell numbers, so it could be confirmed that QS was inhibited instead of growth inhibition. The results are given as the mean ± SD of two parallel measurements.

### 2.9. Statistical Analysis

Statistical analyses were carried out using ExcelR (Microsoft Corp., Redmond, WA, USA) and the PAST software package version 3.11 [48] at a 5% significance level (*p* < 0.05), after normality checking with the Shapiro–Wilk test. Data were expressed as means and standard deviations (SD). The comparisons of quantitative variables between the two groups (honey types) were conducted with one-way ANOVA and Mann–Whitney pairwise comparison. If the null hypothesis of the ANOVA was rejected, we used a Student’s t-test to establish a difference between the two groups.

## 3. Results

### 3.1. Pollen Analysis and Sensory Characteristics of Honey Samples

Identification of honey samples based on their sensory characteristics and melissopalynological analysis revealed that each sample was a unifloral honey (Table 1 and Table 2). The pale color and liquid consistency of the black locust (acacia) honey sample, as well as the high percentage of *Robinia* pollen, confirmed that its botanical source was the flowers of *Robinia pseudoacacia*. Chestnut honey was characterized by dark amber color and liquid consistency and met the requirement of strongly over-represented *Castanea* pollen. In the amber-colored, semisolid goldenrod honey sample, *Solidago* pollen was determined as the dominant pollen type, while Asteraceae and Brassicaceae pollen were important minor pollen. The dominant pollen of the light amber, semisolid linden honey was *Tilia*, confirming its unifloral origin. Milkweed honey was treated as unifloral honey, even though it did not contain any *Asclepias* pollen, but *Brassica* pollen was identified as the dominant pollen type. This can be explained by the fact that the pollen grains of *Asclepias* are dispersed in large units called pollinia, which cannot be collected by honeybees and thus do not enter the honey. The sensory traits of our milkweed honey sample were in accordance with what was expected in the case of this honey type.

### 3.2. MIC Determination

The MIC values are summarized in Table 3. Our results showed that the darker chestnut and linden honey samples were more active than the light-colored honeys, such as black locust and milkweed. In the case of chestnut and linden samples, the MIC values were between 10 and 12.5%; however, the MIC was 20 to 25% when treating the bacterial strains with black locust, goldenrod and milkweed honey samples. The most resistant pathogen was *P. aeruginosa*, which required the highest concentrations of honey to inhibit growth (Table 3).

### 3.3. Antibiofilm Capacity and Cell Viability

The antibiofilm activity of chestnut and linden honey was the most remarkable, inhibiting the most sensitive *S. epidermidis* by 71.1 and 68.7%, respectively. The biofilms of *P. aeruginosa* and MRSA were more resistant than that of *S. epidermidis*. The biofilm formation of *P. aeruginosa* was inhibited by 49.2 to 66.0 and 68.2% by the least active milkweed honey and the most effective linden and chestnut honeys, respectively. Similarly, in the case of MRSA, linden and chestnut honeys were the most effective, with inhibitory rates of 63.2 and 66.9%, respectively. Milkweed and black locust honey showed the lowest activity for each bacterium (Figure 1).

Our cell viability assay revealed that there were viable bacterium cells in the biofilm besides the dead cells. In the case of all three bacteria, the percentage of living cells was over 50%. The greatest reduction in the number of viable cells was observed in the case of chestnut honey (Table 4).

### 3.4. Scanning Electron Microscopy (SEM)

The SEM images illustrate the inhibitory effect of chestnut honey on biofilm formation. SEM tests were performed with this honey type because in our antibiofilm assays chestnut honey proved to be the most active. In the case of untreated samples, the formation of bacterial biofilms and the appearance of a three-dimensional structure can be observed (Figure 2A,C). In the case of samples treated with chestnut honey, no bacterial biofilm was formed (Figure 2B,D). It is worth mentioning that biofilm formation started in the case of *P. aeruginosa*; however, as can be seen in the picture, the process stopped due to the treatment (Figure 2C,D).

### 3.5. Membrane Degradation

From the tested concentrations of honey solutions (20, 40, 60, 90%), loss of integrity of the bacterial membrane was observed at concentrations of 40% and above, but no membrane degradation was found at 20% (Table 5). Chestnut and linden honey samples resulted in high membrane degradation; at a 60% concentration of these honeys, the lysis of bacterial cells reached up to 39.2% and 44.5%, respectively. The performance of black locust, goldenrod and milkweed honeys was weaker; the lowest activity was detected for milkweed honey (Table 5 and Table 6). To investigate the kinetics of the effect of honey samples, their 60% solutions were measured at different time intervals (20, 40, 60 min). This experiment revealed that bacterial membrane degradation started after 40 min (Table 6). Furthermore, our results showed that Gram-positive bacteria were more resistant than Gram-negative *P. aeruginosa* (Table 5 and Table 6).

### 3.6. Anti-Quorum Sensing Effect

Based on the liquid culture assays performed, all honey samples exhibited anti-QS activity (Figure 3). Chestnut had a very strong growth inhibitory effect; no cell numbers could be detected between 75% (*v*/*v*) and 5% (*v*/*v*) concentrations. Between 2–0.5% (*v*/*v*) cell numbers increased to log 6–log 8 CFU/mL, but no pigment production could be detected, indicating the anti-QS effect of this type of honey. Black locust, goldenrod, linden and milkweed honey totally reduced cell growth between 75–25% (*v*/*v*) concentrations. QS inhibition was observed from a 10% honey concentration, where cell numbers reached log 5–log 7 CFU/mL, respectively, but pigment production was still inhibited compared to the control samples. At the lowest concentration tested (0.5% (*v*/*v*)) neither of the samples reached the CFU/mL or OD value of the control, indicating that both growth inhibition and anti-QS effects took place.

## 4. Discussion

Our previous research demonstrated the antibacterial and antibiofilm effects of Central European unifloral honeys against food-borne pathogens and respiratory tract bacteria [49,50]. The present study confirmed that black locust (acacia), chestnut, goldenrod, linden and milkweed honeys effectively hinder the growth, biofilm formation and QS mechanisms of wound-associated bacteria, and thus can serve as valuable alternatives to antibiotics in topical therapies. The novelty of our research is that it reports on the effects of both common and less well-known honey types against bacteria in chronic wounds. This is the first study to investigate the antibacterial and biofilm-inhibiting effects of some specialty honeys, such as goldenrod and milkweed honeys. In addition, our work sheds some light on the mechanisms of action behind honey’s antibiofilm activity, including the demonstration of the anti-quorum sensing effect of varietal honeys. All honey types were good anti-QS agents even at low concentrations (0.5% *v*/*v*), chestnut being the strongest inhibitor. Based on these results, without a growth inhibitory effect, an anti-QS effect was detectable, which indicates that the honey samples tested intervened in the biosynthesis of AHL signal molecules.

Our results revealed that the most significant antibacterial activity was shown by the darker-colored chestnut and linden honeys, in accordance with Albaridi (2019) [51]. This is also in agreement with the work of Truchado et al. [52], who tested 29 honey samples and found that each honey type was capable of interfering with QS, but chestnut and linden honeys had the highest anti-QS activity [52]. In accordance with our findings, Oliveira and coworkers [53] proved the antibiofilm effect of 50% chestnut honey against *Escherichia coli* and *P. aeruginosa* in connection with wound healing. In our study, a much lower concentration (6.25%) of chestnut honey inhibited the biofilm of *P. aeruginosa,* with a 68% inhibitory rate. In a recent study by Sevin and Yarsan [54], the efficiency of chestnut honey cream was proven in a rat wound model where inflammation, as well as levels of granulation tissue, were significantly reduced by the honey treatment compared to the control.

Similarly, linden honeys from Central and Northern Europe were found to exhibit strong antibacterial and antibiofilm activity [49,50,55,56]. In a study analyzing the effect of eleven Danish honeys and manuka honey, linden honey exhibited an even stronger antibacterial effect than manuka honey [55]. The analysis of five Slovakian honey types by Farkasovska et al. [56] revealed that linden honey samples showed the strongest antibacterial activity against *S. aureus* and *P. aeruginosa*, compared to acacia, rapeseed, sunflower and multifloral honey samples. The antibacterial activity of most honey types was found to be mainly H_2_O_2_-dependent [30,56]; however, the strong antibacterial effect of linden honey seems to be at least partially attributed to non-peroxide factor(s) [56]. Sakač and co-workers [57] also observed the outstanding antibacterial effect of chestnut and linden honeys from the Balkan region against *S. aureus* and *S. epidermidis*, with 3.12 to 12.5% and 6.25 to 12.5% MIC values, respectively, which were in a similar range to those of our linden and chestnut honeys, their MIC being 10% against *S. epidermidis*. Differences in MIC values measured for the same type of honey in different countries or even in different regions of the same country can be due to differences in the actual composition of honey types, which in turn can be influenced by the year of harvest and geographical origin.

Although in our experiments black locust/acacia honey displayed weaker activity against wound-associated bacteria, in the study of Ranzato et al. [58] acacia and buckwheat honeys were more active in promoting scratch wound closure compared to manuka honey samples. Cytokines are known to play a role in wound healing. Ranzato et al. [58] reported that acacia honey induced significant increases of primary interleukins present in the fibroblast culture and pointed out a correlation between interleukin modulation and wound healing activity.

Our study was the first to report on the antibacterial, antibiofilm and anti-QS activity of goldenrod and milkweed honeys against the bacteria *S. epidermidis* and MRSA. The antibacterial effect of goldenrod honey was investigated against *P. aeruginosa* and *S. aureus* [49], but no previous research has been dedicated to the wound-healing potential of this honey type. The present study revealed that goldenrod honey had a medium level of antibacterial and antibiofilm activity against wound-associated pathogens, which was lower than that of chestnut and linden honeys but higher than that of acacia and milkweed honeys. As for milkweed honey, only our previous study [49] reported its ability to hinder the growth and biofilm formation of *P. aeruginosa* and *S. aureus*, but this honey type showed the lowest antimicrobial activity in each assay from a set of four Hungarian honeys. Similarly, in the present study, we found that milkweed honey had the lowest or second lowest (after acacia honey) antibacterial potential in most assays against wound-associated bacteria.

The outcomes of the current study, performed on wound-infecting bacteria, confirmed the high antibacterial and antibiofilm potential of linden and chestnut honeys, which were demonstrated previously against the respiratory tract and food-borne pathogens [49,50]. Regardless of the year when our honey samples were harvested, honey types of the same botanical origin inhibited the growth and biofilm formation of various bacteria in a similar manner. Similarly, Truchado et al. [52] concluded that the floral origin of honeys was the most decisive factor regarding QS inhibitory activity, independent of geographic location. In addition, they found that unifloral honey samples showed ‘‘non-peroxide” anti-QS activity, which was not linearly correlated with total and individual phenolic compounds. Our previous results also indicated that linden honey, exhibiting high levels of antibacterial activity in each assay, had a lower total phenolic content/total reducing capacity [49,59] compared to other unifloral honeys with weaker antibacterial properties. Similarly, a study of 12 honey types from the Balkan region reported that linden honey was one of the strongest antibacterial agents against *Staphylococcus* strains, but its polyphenol levels were significantly lower than those of similarly effective phacelia honey [57]. Besides the marker compound lindenin, high levels of methyl syringate were measured in linden honey [60]. The latter phenolic compound is known to act as a potent antioxidant and antibacterial agent [61], and may thus contribute to the exceptional antibacterial properties of linden honey. In chestnut honey, kynurenic acid was identified as a specific compound with antioxidant and antibacterial potential; however, it did not reach a concentration that could justify its high level of antibacterial activity [62]. Further investigation is needed to determine which other unique compounds are associated with the high antibacterial and antibiofilm potential of chestnut and linden honeys.

Not all bacterial strains tested reacted in the same manner to the honey treatments. In our study, the most sensitive pathogen was *S. epidermidis*, which required the lowest concentration of honey to inhibit growth. Additionally, the biofilms of MRSA and *P. aeruginosa* were more resistant than that of *S. epidermidis,* being inhibited by 66.9%, 68.2% and 71.1% by the most effective chestnut honey, respectively. In contrast, when treated with sidr and manuka honeys, *P. aeruginosa* biofilms were more sensitive compared to both methicillin-susceptible and -resistant *S. aureus* strains [63]. At the same time, in our membrane degradation experiments, Gram-positive *Staphylococcus* strains were more resistant than Gram-negative *P. aeruginosa*. Similar observations were reported for avocado, chestnut and manuka honeys, which were more effective against Gram-negative *E. coli* than Gram-positive *S. aureus* [64,65]. A possible explanation for this difference is that honey increases the permeability of the outer membrane of Gram-negative bacteria by destroying the lipopolysaccharide layer [66].

## 5. Conclusions

Honey has great potential as a topical agent in the treatment of wounds due to its antimicrobial, anti-inflammatory and antioxidant properties. It has to be highlighted that different honey types display different levels of health benefits. From a set of five unifloral Hungarian honeys tested against wound-associated bacteria, the antibacterial, biofilm-inhibiting and anti-QS activities of linden and chestnut honeys were superior to those of acacia, goldenrod and milkweed honeys. In addition to the floral origin, the antibacterial effect of honey is also affected by the microbial species treated. Honey can play a significant role as a supplement to antibiotic therapy because it achieves its antibacterial effect at several points of attack.

## Figures and Tables

**Figure 1 microorganisms-11-00509-f001:**
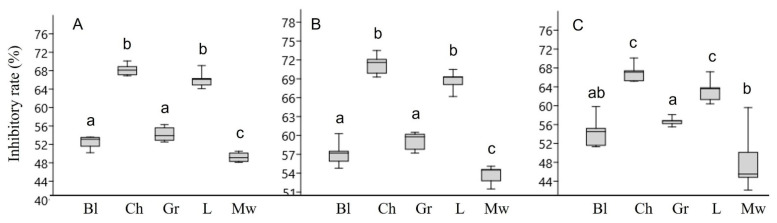
Biofilm inhibitory rates of honey samples against *P. aeruginosa* (**A**), *S. epidermidis* (**B**), MRSA: methicillin-resistant *S. aureus* (**C**). Bl-black locust honey, Ch-chestnut honey, Gr-goldenrod honey, L-linden honey, Mw-milkweed honey. Different lower case letters above the boxes indicate significant differences among various honeys according to Student’s *t*-test (*p* < 0.01).

**Figure 2 microorganisms-11-00509-f002:**
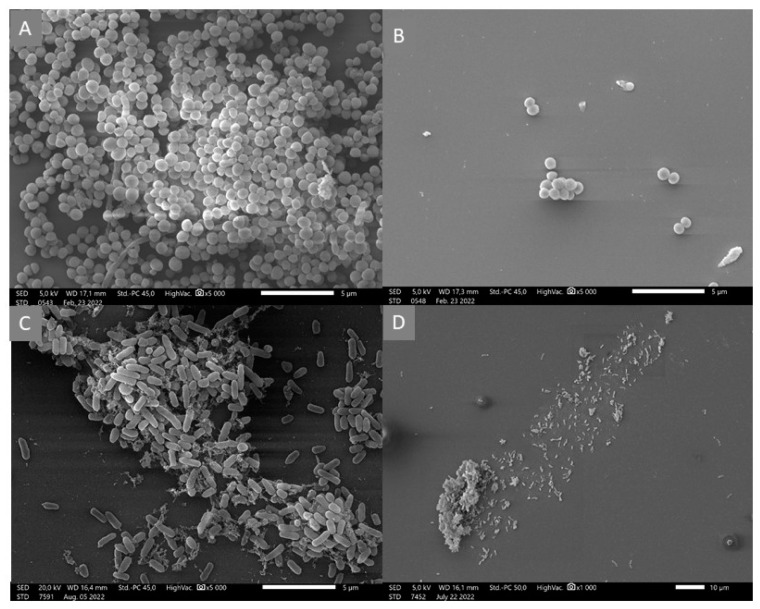
Scanning electron microscopic images of *S. epidermidis* (**A**,**B**) and *P. aeruginosa* (**C**,**D**) Control samples of bacterial strains (**A**,**C**); treatment with MIC/2 concentrations of chestnut honey in the case of *S. epidermidis* and *P. aeruginosa* (**B**,**D**).

**Figure 3 microorganisms-11-00509-f003:**
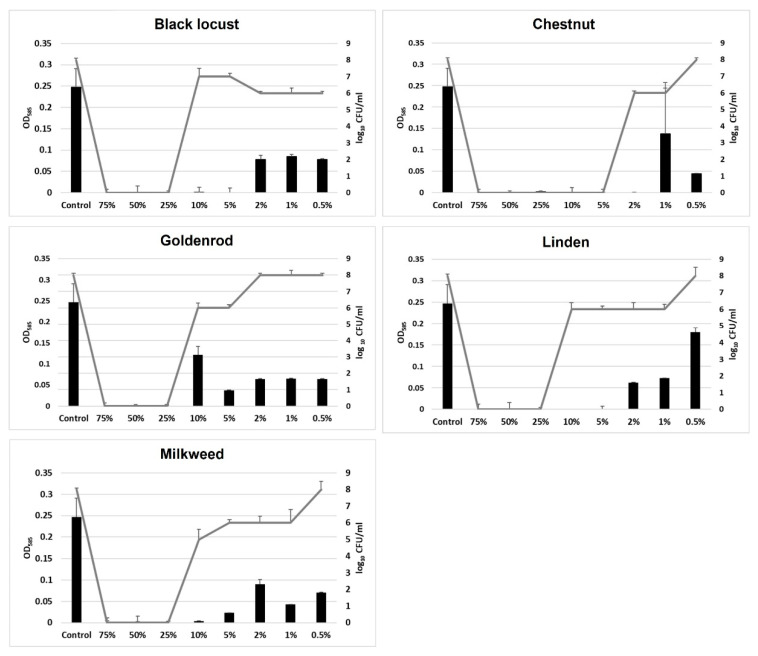
Anti-quorum sensing activity of unifloral honey samples. Bars represent OD values at 585 nm and lines represent cell numbers defined in log10 CFU/mL. Data are given as mean ± SD.

**Table 1 microorganisms-11-00509-t001:** Sensory characteristics and color of the analyzed honey samples.

Nr.	Honey Type, Plant Name	Sensory Characteristics (Color, Odor and Consistency)	ABS_450–720_ (mAU)
1	Black locust, *Robinia pseudoacacia*	Pale, yellowish green, weak odor, liquid, viscous	83.3 ± 14.2 ^a^
2	Chestnut, *Castanea sativa*	Amber with reddish tone, strong odor, liquid, viscous	413.0 ± 21.2 ^b^
3	Goldenrod, *Solidago gigantea*	Amber, moderately intense odor, semisolid, fine granulated	412.3 ± 13.0 ^b^
4	Linden, *Tilia* spp.	Light amber, strong odor, fine granulated, semisolid	284.0 ± 9.2 ^c^
5	Milkweed, *Asclepias syriaca*	Light yellowish amber, intense flower-like odor, viscous	119.0 ± 17.8 ^d^

ABS_450–720_: absorbance of diluted honey samples referring to their color. Each code number in the first column represents three biological replicates (n = 3) of the honey samples. Data in the last column with different superscripted letters mean significant differences among various honeys according to the Student’s *t*-test (*p* < 0.01).

**Table 2 microorganisms-11-00509-t002:** Pollen spectrum of the studied honeys.

Honey Type	*Castanea*	*Robinia*	*Solidago*	*Tilia*	Asteraceae	*Brassica*	Other
black locust	-	64.2	11.2	0.4	1.9	6.8	15.5
chestnut	79.9	-	0.8	3.2	2.8	3.1	10.2
goldenrod	-	-	72.4	1.9	8.4	6.7	10.6
linden	-	1.9	19.5	72.4	2.7	3.2	0.3
milkweed	-	1.5	-	-	0.9	65.4	32.2

Dominant pollen >45%, secondary pollen 16–45%, important minor pollen 3–15%, minor pollen <3% of the pollen grains counted.

**Table 3 microorganisms-11-00509-t003:** The MIC value of black locust, chestnut, goldenrod, linden and milkweed honey.

	Honey Samples	1	2	3
	black locust	25%	20%	20%
	chestnut	12.5%	10%	10%
MIC values	goldenrod	25%	20%	20%
	linden	12.5%	10%	10%
	milkweed	25%	20%	20%

1: *P. aeruginosa*, 2: *S. epidermidis*, 3: MRSA: methicillin-resistant *S. aureus.* Percentage values of the table correspond to dilution % of honey causing antimicrobial effects.

**Table 4 microorganisms-11-00509-t004:** Cell viability of *S. epidermidis*, MRSA and *P. aeruginosa* treated by honey samples.

Percentage of Viable Cells (%)			
Honey Samples	1	2	3
black locust	57.4	62.7	52.7
chestnut	56.7	50.4	52.3
goldenrod	67.4	53.7	56.7
linden	61.1	61.2	62.8
milkweed	72.1	75.2	79.4

1: *P. aeruginosa*, 2: *S. epidermidis*, 3: MRSA.

**Table 5 microorganisms-11-00509-t005:** Effect of honey solutions at different concentrations on the release of cellular material from Gram+ (*S. epidermidis*, MRSA) and Gram− (*P. aeruginosa*) bacteria.

Conc (%)	Lysis of *S. epidermidis* cells				
	Black Locust	Chestnut	Goldenrod	Linden	Milkweed
0	0	0	0	0	0
20	0	0	0	0	0
40	17.45 ± 3.1 ^a^	22.35 ± 2.0 ^bc^	20.87 ± 2.2 ^ac^	30.72 ± 1.6 ^d^	16.88 ± 2.0 ^a^
60	25.92 ± 2.9 ^a^	30.68 ± 1.9 ^b^	32.05 ± 2.2 ^b^	38.30 ± 4.2 ^d^	24.08 ± 2.9 ^a^
90	100	100	100	100	100
	**Lysis of MRSA cells**				
0	0	0	0	0	0
20	0	0	0	0	0
40	21.38 ± 1.6 ^a^	23.37 ± 2.5 ^a^	22.18 ± 4.3 ^a^	30.42 ± 1.8 ^b^	16.35 ± 1.5 ^c^
60	30.22 ± 1.5 ^a^	35.45 ± 2.9 ^b^	33.82 ± 2.6 ^b^	39.38 ± 2.1 ^c^	25.13 ± 3.3 ^d^
90	100	100	100	100	100
	**Lysis of *P. aeruginosa* cells**				
0	0	0	0	0	0
20	0	0	0	0	0
40	21.55 ± 2.2 ^a^	26.80 ± 4.5 ^b^	22.23 ± 4.1 ^ac^	35.77 ± 3.8 ^d^	17.60 ± 3.3 ^c^
60	32.72 ± 4.1 ^a^	39.15 ± 1.3 ^b^	36.30 ± 4.2 ^ab^	44.53 ± 3.9 ^c^	25.48 ± 3.1 ^d^
90	100	100	100	100	100

Different lower case letters in the same row for each bacterium indicate significant differences among various honeys according to Student’s *t*-test (*p* < 0.05).

**Table 6 microorganisms-11-00509-t006:** Kinetics of 260 nm absorbing material released from Gram+ (*S. epidermidis*, MRSA) and Gram− (*P. aeruginosa*) bacteria treated with 60% (*w*/*v*) honey solutions.

Time (Min)	Lysis of *S. epidermidis* cells				
	Black Locust	Chestnut	Goldenrod	Linden	Milkweed
0	0	0	0	0	0
20	0	0	0	0	0
40	21.37 ± 1.4 ^a^	24.60 ± 3.2 ^b^	22.43 ± 2.8 ^ab^	27.07 ± 6.4 ^ab^	15.15 ± 3.3 ^c^
60	25.92 ± 2.9 ^a^	30.68 ± 1.9 ^b^	32.05 ± 2.2 ^b^	38.30 ± 4.2 ^c^	24.08 ± 2.9 ^a^
90	62.63 ± 3.0 ^a^	64.85 ± 4.0 ^ac^	54.85 ± 4.2 ^b^	69.08 ± 4.4 ^c^	35.28 ± 6.2 ^d^
	**Lysis of MRSA cells**				
0	0	0	0	0	0
20	0	0	0	0	0
40	27.82 ± 1.9 ^a^	29.95 ± 1.9 ^a^	22.25 ± 3.6 ^b^	32.28 ± 6.8 ^a^	16.50 ± 3.3 ^c^
60	30.22 ± 1.5 ^a^	35.45 ± 2.9 ^b^	33.82 ± 2.6 ^b^	39.38 ± 2.1 ^c^	25.13 ± 3.3 ^d^
90	61.43 ± 4.8 ^a^	65.70 ± 7.7 ^ac^	52.48 ± 4.0 ^b^	71.28 ± 1.1 ^c^	33.87 ± 5.6 ^d^
	**Lysis of *P. aeruginosa* cells**				
0	0	0	0	0	0
20	0	0	0	0	0
40	16.58 ± 3.2 ^a^	26.08 ± 4.0 ^b^	18.18 ± 2.3 ^a^	26.40 ± 2.8 ^b^	11.45 ± 1.3 ^c^
60	32.72 ± 4.1 ^a^	39.15 ± 1.3 ^b^	36.30 ± 4.2 ^ab^	44.53 ± 3.9 ^c^	25.48 ± 3.1 ^d^
90	63.22 ± 3.7 ^a^	70.38 ± 5.6 ^b^	57.67 ± 3.4 ^c^	74.15 ± 3.9 ^b^	43.38 ± 7.0 ^d^

Different lower case letters in the same row indicate significant differences among various honeys according to Student’s *t*-test (*p* < 0.05).

## Data Availability

Not applicable.

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
