# Peer review of "Antibacterial and Antibiofilm Effect of Unifloral Honeys against Bacteria Isolated from Chronic Wound Infections"

_microorganisms, 2023, doi:10.3390/microorganisms11020509_

Round 1

Reviewer 1 Report

Authors of submitted ms focused on antibacterial and antibiofilm effect of several types of honey from Hungary. Overall, this study lacks of novelty and actually does not bring any new information to the current state of knowledge.

My major concern is about the methodology:

1. I appreciate that author performed the pollen analysis of honey tested in the study, however, basic physico-chemical characterisitcs would also be great to see.

2. No individual antibacterial compounds in tested honey was analysed. The content of hydrogen peroxide or defensin-1, or determination of glucose oxidase activity is important especially if authors tested new honey types which have not been investigated yet.

3. Why did the authors use BHI cultivation medium instead of MHB medium which is recommended medium for testing of antibacterial compounds by NCCLS guidelines. The description of the method should be done in more detail, for example shaking and time conditions for microplate, final concentration of bacteria in the well etc.

4. Approach to determine antibiofilm activity needs to be updated. What is the viability of bacteria embedded within the formed biofilm. Authors need determine the viable bacteria in biofilm after treatment with honey and compare the overall count with control.

5. Effect of honey on bacteria embdedded in monobacterial biofilm is well known and well described. Authors may focus on mixed biofilm model with various bacteria and bring some new approach.

6. Discussion is rather poor and simple. I strongly suggest to compare your obtained results with other studies and for example explain why MIC values of tested honeys are pretty high in compared to other studies. It is because of differences in cultivation media or different methodology? MIC values of tested honeys are almost 10-times higher that in other studies with similar honey types and bacteria.

Overall, ms is weak and need substantial revision with new experimental data.

Author Response

Answers to Reviewer 1

Authors of submitted ms focused on antibacterial and antibiofilm effect of several types of honey from Hungary. Overall, this study lacks of novelty and actually does not bring any new information to the current state of knowledge.

My major concern is about the methodology:

  1. I appreciate that author performed the pollen analysis of honey tested in the study, however, basic physico-chemical characterisitcs would also be great to see.

We agree that physico-chemical characteristics are important for the complex characterisation of honeys. However, melissopalynological analysis combined with spectrophotometric colour determination is also accepted as the means of confirming the botanical origin of honey samples (Kús et al. 2014, Escuredo et al. 2019), which was our primary aim with performing these examinations. In addition, sensory characteristics are presented in the MS (revised MS, Table 1).

References:

Escuredo, O.; Rodríguez-Flores, M.S.; Rojo-Martínez, S.; Seijo, M.C. Contribution to the chromatic characterization of unifloral honeys from Galicia (NW Spain). Foods 2019, 8, 233.

Kús, P.M.; Congiu, F.; Teper, D.; Sroka, Z.; Jerkovic, I.; Tuberoso, C.I.G. Antioxidant activity, color characteristics, total phenol content and general HPLC fingerprints of six Polish unifloral honey types. LWT Food Sci. Technol. 2014, 55, 124–130.

  1. No individual antibacterial compounds in tested honey was analysed. The content of hydrogen peroxide or defensin-1, or determination of glucose oxidase activity is important especially if authors tested new honey types which have not been investigated yet.

We acknowledge that the analysis of compounds that can be responsible for the antibacterial effect of honeys is important. In fact, such analyses - including the determination of polyphenol content and composition in honey samples - are currently running in our laboratory. We would like to summarize the results of these analyses in a separate publication.

The hydrogen peroxide and/or glucose oxidase levels of some honey types included in our investigation were already determined: for acacia, chestnut, and linden honey samples by Bucekova et al. (2019), Cviljevic et al. (2020), and Farkasovska et al. (2019), respectively. In goldenrod honey ethyl 4-ethoxybenzoate and 9-methylphenanthrene were identified as antibacterial compounds (Wright et al. 2020).

References:

Bucekova, M.; Jardekova, L.; Juricova, V.; Bugarova, V.; Di Marco, G.; Gismondi, A.; Leonardi, D.; Farkasovska, J.; Godocikova, J.; Laho, M. Antibacterial Activity of Different Blossom Honeys: New Findings. Molecules 2019, 24, 1573.

Cviljević, Sabina; Bilić Rajs, Blanka; Primorac, Ljiljana; Strelec, Ivica; Gal, Katarina; Cvijetić Stokanović, Milica; Penava, Ariana; Mindum, Anita; Flanjak, Ivana. Antibacterial activity of chestnut honey (Castanea sativa Mill.) against Helicobacter pylori and correlation to its antioxidant capacity // Hrana u zdravlju i bolesti, 9 (2020), 2; 52-56.

Farkasovska, J.; Bugarova, V.; Godocikova, J.; Majtan, V.; Majtan, J. The Role of Hydrogen Peroxide in the Antibacterial Activity of Different Floral Honeys. European Food Research and Technology 2019, 245, 2739–2744.

Wright, J.; Masown, K.; Ruffel, S.E. Comparison of antibacterial Apis mellifera honey varietals from northeastern United States. BIOS 2020, 91 (3); 180-187.

  1. Why did the authors use BHI cultivation medium instead of MHB medium which is recommended medium for testing of antibacterial compounds by NCCLS guidelines. The description of the method should be done in more detail, for example shaking and time conditions for microplate, final concentration of bacteria in the well etc.

Based on previous studies, we deemed it appropriate to use the BHI. Since BHI better supports the growth of pathogenic bacteria like pneumococci and staphylococci as compared to MHB, which is traditionally used for susceptibility studies, we assessed our measurements in this media (Parvenkar et al., 2020; Vestergaard et al., 2021).

Shaking and time conditions are included in 2.3. Cultivation of Test Bacteria. Final concentration of bacteria in wells was 105 CFU/ml in case of MIC determination, 108 CFU/ml in case of antibiofilm determination, which data are included in Material and Methods (2.4. and 2.5., respectively).

References:

Parvenkar P., Palaskar J., Metgud S., Maria R., Dutta S. (2020): The minimum inhibitory concentration (MIC) and minimum bactericidal concentration (MBC) of silver nanoparticles against Staphylococcus aureus. Biomater Investig Dent. 7(1): 105–109. 10.1080/26415275.2020.1796674

Vestergaar M., Skive B., Domraceva I., Ingmer H., Franzyk H. (2021): 4.3. Determination of Minimum Inhibitory Concentration (MIC) and Minimum Bactericidal Concentration (MBC). Int J Mol Sci. 10.3390/ijms22115617

  1. Approach to determine antibiofilm activity needs to be updated. What is the viability of bacteria embedded within the formed biofilm. Authors need determine the viable bacteria in biofilm after treatment with honey and compare the overall count with control.

The bacterial biofilm contains not only viable bacterium cells, but dead cells as well. During the biofilm formation process, there are bacterial cells which release their cellular material, in order to provide a basis to the other cells (Høiby, 2014; Dufour et al., 2012). With CV assay it is possible to measure the whole biomass of biofilm, that was the reason why it was chosen. However, we made the cell viability test as well, which is now included in the manuscript (2.5. and 3.3.).

References:

Høiby N. A. (2014): Personal history of research on microbial biofilms and biofilm infections. Pathogens Dis. 70:205-211.

Dufour D., Leung V., Lévesque C. (2012): Bacterial biofilm: structure, function, and antimicrobial resistance. Endodontic Topics. 22:2-16.

  1. Effect of honey on bacteria embedded in monobacterial biofilm is well known and well described. Authors may focus on mixed biofilm model with various bacteria and bring some new approach.

We thank the Reviewer for his comment. We definitely plan to investigate mixed-culture biofilms as well, due to the fact that this form of biofilm is more common than monobacterial forms. The aim of the present study was to determine if our honey samples have any effect on monobacterial biofilms. Moving on with our studies and using the obtained results with monobacterial biofilms, the mixture of bacteria will be the next step. In fact, as a pilot study we already carried out a SEM study with mixed biofilm, the results of which are demonstrated below:

Pseudomonas aeruginosa and MRSA biofilms observed using scanning electron microscope. A: control of mixed biofilm, B: treated biofilm with MIC/2 concentration of linden honey

  1. Discussion is rather poor and simple. I strongly suggest to compare your obtained results with other studies and for example explain why MIC values of tested honeys are pretty high in compared to other studies. It is because of differences in cultivation media or different methodology? MIC values of tested honeys are almost 10-times higher that in other studies with similar honey types and bacteria.

Comparing our results with those of other research groups is in fact crucial. The original version of our manuscript already includes such comparisons, e.g. we mentioned that similar to our findings, Oliveira and co-workers (2018) reported the antibiofilm effect of 50% chestnut honey against wound-associated bacteria. The MIC values obtained for our linden honey samples were contrasted with those reported by Sakač and co-workers (2022).

The revised version of our manuscript includes further comparative evaluation of our results, and a short explanation for differences in MIC values was added - which is also detailed below.

The minimum inhibitory concentrations (MIC) were determined with broth microdilution method. Microtiter plates (96-wells) were used to perform this assay. From each bacterium solution (105 cfu/mL) 100 μL was measured to the wells. From honey samples 20%, 30%, 40%, 50%, 60% and 70% (w/v) stock solutions were prepared. The honey samples were solved in BHI, from these solutions 100 μL was added to the treated wells. It was taken into consideration that during the assay the concentration of honey solutions was halved. Preparing the solutions between 20-50%, the honey samples were diluted in BHI, based on the method mentioned above.  In case of 60 and 70% solutions, the bacterial suspension was used to dilute the honey samples. The amount of honey and BHI solution needed to achieve the appropriate concentrations are provided in Table 1 below. After incubation (24 h, 37°C) absorbance was measured at 600 nm with a microtiter plate reader (BMG Labtech, SPECTROstar Nano, Budapest, Hungary). The negative control was the clear BHI, the positive control was the untreated bacterial suspension. The average of the six replicates was calculated and then the mean of the negative control was subtracted from the value obtained. Absorbance lower than 10% of the positive control samples, i.e. growth inhibition of 90% or more, was considered as the MIC value.

Table 1. Calculating of concentration of honey samples

Concentration of treating solutions

Amount of honey (g)

Amount of BHI (µL)

20%

0.8

1200

30%

1.2

800

40%

1.6

400

50%

2.0

0

treating solutions

Amount of honey (g)

Amount of bacterial suspension (µL)

60%

1.2

800

70%

1.4

600

Differences in MIC values can be due to the composition of the honey, the different vintage, the different geographical origin. This work also highlights the fact that it is difficult to standardize honeys, each vintage and region can differ and the individual sensitivity of the bees and the nutrient solution/breeding conditions must also be taken into account.

Reviewer 2 Report

Dear Dr.

Editor of Microorganisms

Authors Thank you very much for choosing me for reviewing to your esteemed Journal (Microorganisms).

Please find the comments.

Abstract:

 It should include background, objective, materials and methods, results, and a brief conclusion.

Introduction:

· A paragraph on the physicochemical properties of honey needs to add and through the light about the factors that affect them like botanical origin, environment, geographical origin, bee species/subspecies,……etc.

- Taha, E-K.A., Al-Kahtani, S.N.; Taha, R. (2021). Comparison of the physicochemical characteristics of sidr (Ziziphus spp.) honey produced by Apis florea F. and Apis mellifera L. Journal of Apicultural Research, 60(3): 470–477.

Materials and Methods

·       Line 135: please delete the year from the reference.

·       Statistical analysis: What is the type of experimental design?

·       What is the ANOVA type used in this study?

Results

·       In Fig. 2, letters above the bars should be added.

·       The standard error in the chestnut column (1%) seems close to the mean value, I think something wrong.

Discussion

·       A paragraph on pollen analysis of honey samples needs to add. The authors can utilize with

- Taha, E-K.A., Al-Kahtani, S.N.; Taha, R. (2018). Comparison of pollen spectra and amount of mineral content in honey produced by Apis florea F. and Apis mellifera L. Journal of the Kansas Entomological Society, 91(1): 51-58.

-  Taha, E-K.A., Al-Kahtani, S.N.; Taha, R. (2021). Comparison of the physicochemical characteristics of sidr (Ziziphus spp.) honey produced by Apis florea F. and Apis mellifera L. Journal of Apicultural Research, 60(3): 470–477.

·       Line 323, 326, 328, 336, 341, 345, 348, and 368: please delete the year from the references.

Author Response

Answers to Reviewer 2

Abstract:

 It should include background, objective, materials and methods, results, and a brief conclusion.

In fact, the abstract includes all the above sections, but without the appropriate headings (as requested by the journal.

Introduction:

A paragraph on the physicochemical properties of honey needs to add and through the light about the factors that affect them like botanical origin, environment, geographical origin, bee species/subspecies,……etc.

- Taha, E-K.A., Al-Kahtani, S.N.; Taha, R. (2021). Comparison of the physicochemical characteristics of sidr (Ziziphus spp.) honey produced by Apis florea F. and Apis mellifera L. Journal of Apicultural Research, 60(3): 470–477.

Thank you for calling our attention to the importance of physicochemical properties of honey and the factors that may influence these traits. In the revised version of the manuscript we included information on this aspect, as well, in addition to the botanical and geographical origin of honey, which were already part of the original manuscript.

Materials and Methods

  • Line 135: please delete the year from the reference.

Thank you for your remark, the year has been deleted.

  • Statistical analysis: What is the type of experimental design?

Thank you drawing our attention to the inaccurate statistical description. It has been clarified with the details below, and unnecessary parts have been deleted.

The honey types had been compared with each other based on a given parameter
using ANOVA. If the null hypothesis of the ANOVA is rejected, we used a Student t-test to establish a difference between the two groups (honey types).

What is the ANOVA type used in this study?

One-way ANOVA, Mann-Whitney pairwise comparison was used.

Results

  • In Fig. 2, letters above the bars should be added.

As mentioned in 2.8., two parallel measurements were carried out, which was not sufficient to perform ANOVA. Thus, results are presented only as mean±SD.

  • The standard error in the chestnut column (1%) seems close to the mean value, I think something wrong

We have double-checked our data sets and their representation in this figure. In fact, standard deviation had high values in case of 1% chestnut honey solutions.

Discussion

A paragraph on pollen analysis of honey samples needs to add. The authors can utilize with

- Taha, E-K.A., Al-Kahtani, S.N.; Taha, R. (2018). Comparison of pollen spectra and amount of mineral content in honey produced by Apis florea F. and Apis mellifera L. Journal of the Kansas Entomological Society, 91(1): 51-58.

-  Taha, E-K.A., Al-Kahtani, S.N.; Taha, R. (2021). Comparison of the physicochemical characteristics of sidr (Ziziphus spp.) honey produced by Apis florea F. and Apis mellifera L. Journal of Apicultural Research, 60(3): 470–477.

Since the focus of our current study was the antibacterial and antibiofilm potential of honeys, and not the quality assessment of honey types, we did not think it necessary to include a separate paragraph on pollen analysis in Discussion. In this case, sensory evaluation and melissopalynological analysis of our honey samples served only as the means of confirming their botanical origin.

Our previous publications (e.g. Farkas et al. 2022) provide comparative evaluation of pollen spectra and mineral content of various Central European honey types, including the ones that were used in the current study.

References:

Farkas Á, Balázs VL, Kőszegi T, Csepregi R, Kerekes E, Horváth G, Szabó P, Gaál K and Kocsis M (2022) Antibacterial and Biofilm Degradation Effects of Hungarian Honeys Linked With Botanical Origin, Antioxidant Capacity and Mineral Content. Front. Nutr. 9:953470. doi: 10.3389/fnut.2022.953470

  • Line 323, 326, 328, 336, 341, 345, 348, and 368: please delete the year from the references.

Years have been deleted.

Reviewer 3 Report

This manuscript describes the antibacterial and antibiofilm effect of unifloral honeys produced in Hungary against bacteria isolated from chronic wound infections. Alhtough the topic is interesting authors should address the following issues:

1.  Why authors did not follow CLSI guidelines to perform MIC determination? It could be easier in that way to compare their results with other similar studies.Is any particular reason to use BHI instead of Muller-Hinton broth? Authors actually did no determine MIC but instead MIC90. They should clarify that point.

2. Line 179: Why bacterial cells where incubated firstly just for 4 h and then for 24 h (in the presence of honey in both cases or just in the second)? Thatr way  the measured activity was neither against biofilm formation nor eradication.. How many were the replicates? Why SD is not mentioned in results??

3. Lines 199-200: Authors should mention that they actually measure the nucleic acids present  at the supernatant . Furthermore centrifugation should be descirbed in g not rpm.

4. Overall, my strongest criticism about this study is the fact that is rather fragmented and at some points not well designed. For instance, there is no direct evidence of QS inhibition  regarding any of the tested  pathogens  in this study. I think it could be more straight forward if authors  could measure virulence factors directly regulated by QS for example LasA protease, LasB elastase , pyoverdine in P. aeruginosa or the expression of las and rhl QS genes ( by Real-time PCR in the presence of honeys). Similarly, membrane destruction could be directly demonstrated by elecron microscopy.

5. Finally it is not clear if most of the data presented in this study are new and not somehow a reptition of previous studies conducted by the same authors (references 39, 40). I think authors should clealry put in perspective and compare their current data with their previous studies regarding same type honeys produced in the same areas but just in different year.

Author Response

Answers to Reviewer 3

This manuscript describes the antibacterial and antibiofilm effect of unifloral honeys produced in Hungary against bacteria isolated from chronic wound infections. Although the topic is interesting authors should address the following issues:

  1. Why authors did not follow CLSI guidelines to perform MIC determination? It could be easier in that way to compare their results with other similar studies. Is any particular reason to use BHI instead of Muller-Hinton broth? Authors actually did no determine MIC but instead MIC90. They should clarify that point.

Based on previous studies, we deemed it appropriate to use the BHI. Since BHI better supports the growth of pathogenic bacteria like pneumococci and staphylococci as compared to MHB, which is traditionally used for susceptibility studies, we assessed our measurements in this media (Parvenkar et al., 2020; Vestergaard et al., 2021).

As for MIC determination, it is correct to use the term MIC in our study. The terms MIC50 and MIC90 are used when we want to know how good an antimicrobial works against a species, and perform MIC testing on a large panel of isolates of that species. The MIC50, which requires testing at least 100 isolates, gives the MIC, which inhibits 50% of the isolates, while the MIC90 (may be the result of different test panels in more than one laboratory) gives the MIC which inhibits 90% of the isolates of the species tested (https://clsi.org/standards/products/iso-documents/documents/iso-20776-1-2019/).

References:

Parvenkar P., Palaskar J., Metgud S., Maria R., Dutta S. (2020): The minimum inhibitory concentration (MIC) and minimum bactericidal concentration (MBC) of silver nanoparticles against Staphylococcus aureus. Biomater Investig Dent. 7(1): 105–109. 10.1080/26415275.2020.1796674

Vestergaar M., Skive B., Domraceva I., Ingmer H., Franzyk H. (2021): 4.3. Determination of Minimum Inhibitory Concentration (MIC) and Minimum Bactericidal Concentration (MBC). Int J Mol Sci. 10.3390/ijms22115617

  1. Line 179: Why bacterial cells where incubated firstly just for 4 h and then for 24 h (in the presence of honey in both cases or just in the second)? That way the measured activity was neither against biofilm formation nor eradication. How many were the replicates? Why SD is not mentioned in results?

First, we used an incubation time of 4 hours, as this is enough time for the bacteria to adhere to the surface. During the 4-hour incubation no honey was added to the media. After that, in order to inhibit the maturation processes of the biofilm, we used a 24-hour incubation in presence of honey.

The measurements were repeated six times.

We will supplement the manuscript with the above information as well.

Although Figure 1. already shows SD values, the text of the revised manuscript presents rates of inhibition in the format mean ± standard deviation.

  1. Lines 199-200: Authors should mention that they actually measure the nucleic acids present at the supernatant. Furthermore centrifugation should be descirbed in g not rpm.

Thank you for your comment, the manuscript has been supplemented with this sentence: "During the test, the nucleic acid in the supernatant was measured."

The protocol was rewritten in g instead of rpm.

  1. Overall, my strongest criticism about this study is the fact that is rather fragmented and at some points not well designed. For instance, there is no direct evidence of QS inhibition  regarding any of the tested  pathogens  in this study. I think it could be more straight forward if authors  could measure virulence factors directly regulated by QS for example LasA protease, LasB elastase , pyoverdine in P. aeruginosa or the expression of las and rhl QS genes ( by Real-time PCR in the presence of honeys). Similarly, membrane destruction could be directly demonstrated by elecron microscopy.

Our aim with the QS measurements was to determine if the honey samples have any anti-QS effect. For this reason, only C. violaceum model organism was used to test this hypothesis. Based on our results it is necessary to continue these measurements in further studies where we plan to measure the virulence factors mentioned by the reviewer.

The manuscript was supplemented with SEM images illustrating the antibiofilm activity.

  1. Finally it is not clear if most of the data presented in this study are new and not somehow a reptition of previous studies conducted by the same authors (references 39, 40). I think authors should clealry put in perspective and compare their current data with their previous studies regarding same type honeys produced in the same areas but just in different year.

Thank you for your comment, the novelty of the manuscript has been supplemented.

Round 2

Reviewer 1 Report

Authors improved the manuscript but there are still some unanswered issues.

1. A huge differences in MIC values when compare presented data to data from other studies are most likely due to the methodological approach rather than differences in honey compositions. There are even 10-fold differences. Therefore, your data are not comparable with other studies where MHB or other cultivation medium was used.

2. Authors should show the data of basic legislative parameters such as HMF, diastase etc.

Author Response

Authors improved the manuscript but there are still some unanswered issues.

  1. A huge differences in MIC values when compare presented data to data from other studies are most likely due to the methodological approach rather than differences in honey compositions. There are even 10-fold differences. Therefore, your data are not comparable with other studies where MHB or other cultivation medium was used.

In order to make our data comparable to those of other research groups, we have performed new experiments to determine the MIC values of our honey samples, using the broth macrodilution method, following CLSI guidelines. In addition, bacteria were grown in Mueller-Hinton broth instead of BHI. Details of the method were provided in 2.4, while the MIC values obtained with the above method are presented in 3.2 and Table 3 in the revised manuscript.

  1. Authors should show the data of basic legislative parameters such as HMF, diastase etc.

We thank you for calling our attention to the importance of these legislative parameters. Unfortunately, physicochemical parameters like electrical conductivity, pH, HMF, diastase activity etc. of our honey samples were not measured in our laboratory. Below we present data that are available from the work of other Hungarian researchers, but it would not be scientifically correct to include these data in our manuscript.

According to directive 1-3-2001/110 of the Codex Alimentarius Hungaricus, electrical conductivity of Hungarian honeys should be below 0.8 mS/cm, except for chestnut and honeydew honeys, where this value should be at least 0.8 mS/cm; moisture content should not exceed 20%; diastase activity should be at least 8 (Schade scale); and HMF content should not exceed 40 mg/kg.

Based on a large-scale Hungarian study (Czipa 2010), the following data are typical for Hungarian acacia, chestnut, linden and milkweed honey types:

Acacia (N=21)

Chestnut (N=5)

Linden

(N=13)

Milkweed (N=5)

moisture content (%)

18.7 ± 0.8

17. 2 ± 0.2

19.2 ± 0.8

19.1 ± 1.3

total sugar content (%)

79.5 ± 0.6

81.1 ± 0.2

79.1 ± 0.8

78.8 ± 0.5

prolin content (mg/kg)

252 ± 38

644 ± 155

698 ± 248

483 ± 114

electrical conductivity (mS/cm)

0.135 ± 0.020

0.584 ± 0.112

0.623 ± 0.071

0.214 ± 0.019

pH

3.5 ± 0.1

4.1 ± 0.4

4.2 ± 0.3

3.3 ± 0.0

HMF content (mg/kg)

9.8 ± 9.3

19.8 ± 11.5

-

22.8 ± 2.4

diastase activity (DN)

16.1 ± 3.1

16.7 ± 3.2

21.8 ± 3.2

10.3 ± 0.9

These data are in accordance with the more recent study of Bodor et al. (2017):

Honey type

Total soluble dry matter (%)

pH

Electrical conductivity (mS/cm)

acacia

82.83 ±0.06

3.52 ± 0.27

0.127 ±0.02

linden

80.30 ±0.10

3.84 ±0.01

0.553 ± 0.02

Although several recent papers dedicated to the microbiological activity of honeys were published without presenting data on the examined honeys’ physicochemical properties (e.g. Bucekova et al. 2019, Bazaid et al. 2022, Hewett et al. 2022), we acknowledge the importance of these factors, and in our future studies we will perform appropriate measurements and include such data in our research papers.

References:

Bazaid, A.S.; Aldarhami, A.; Patel, M.; Adnan, M.; Hamdi, A.; Snoussi, M.; Qanash, H.; Imam, M.; Monjed, M.K.; Khateb, A.M. The Antimicrobial Effects of Saudi Sumra Honey against Drug Resistant Pathogens: Phytochemical Analysis, Antibiofilm, Anti-Quorum Sensing, and Antioxidant Activities. Pharmaceuticals 2022, 15, 1212.

Bodor Zs., Koncz F.A., John-Lewis Z.Z., Kertész I., Gillay Z., Kaszab T., Kovács Z., Benedek Cs. Effect of Heat Treatment on Chemical and Sensory Properties of Honeys. Animal Welfare, Etológia és Tartástechnológia, 2017, 13 (2). pp. 39-48.

Bucekova, M.; Jardekova, L.; Juricova, V.; Bugarova, V.; Di Marco, G.; Gismondi, A.; Leonardi, D.; Farkasovska, J.; Godocikova, J.; Laho, M. Antibacterial Activity of Different Blossom Honeys: New Findings. Molecules 2019, 24, 1573.

Czipa N. Különböző eredetű mézek összehasonlító vizsgálata és a gyártmánykialakítás hatása a minőségre. Doct. dissertation, University of Debrecen, 2010, pp.142

Codex Alimentarius Hungaricus: https://docplayer.hu/115341789-Codex-alimentarius-hungaricus-magyar-elelmiszerkonyv.html

Hewett, S.R.; Crabtrey, S.D.; Dodson, E.E.; Rieth, C.A.; Tarkka, R.M.; Naylor, K. Both Manuka and Non-Manuka Honey Types Inhibit Antibiotic Resistant Wound-Infecting Bacteria. Antibiotics 2022, 11, 1132.

Reviewer 2 Report

The authors have done most comments. The manuscript can be accepted.

Author Response

The authors have done most comments. The manuscript can be accepted.

Many thanks for the acceptance.

Reviewer 3 Report

The authors have tried to improve the manuscript and to answer my comments and they partly did so. However, in my opinion their response is not satisfactory at certain points:

1. Still I do not understant why authors did not follow CLSI quidelines. A major issue in research regarding antimicrobial activity of honey and other honeybee products is that the use of different protocols makes result interpretation rather confusing. MHB perfectly supports MRSA and P. aeruginosa growth. I do not agree that BHI supports better growth.

2. In my view the implemented  protocol used to assess antibiofilm activity is not well designed. Authors allow bacteria to initiate biofilm formation (4 hours incubation) and then they apply honey to a non-mature biofilm. It is not clear neither if they demonstrate inhibition of biofilm formation nor if they demsontrate eradication of an already formed, mature biofilm.

4. Still authors do no provide any data regarding anti-QS activity specifically for MRSA and/or P. aeruginosa. General anti-QS activity of honey using C. violaceum as a model microorganism has been published in several articles (including a very recent one  https://www.mdpi.com/1424-8247/15/10/1212). Unless authors provide novel,  specific data on anti-QS activity, I suggest they remove the anti-QS data

5. Authors mention that the novelty of the manucsript has been supplement. They mean Lines 443-448? Elsewhere? Please mention all points throughout manuscript.

Author Response

The authors have tried to improve the manuscript and to answer my comments and they partly did so. However, in my opinion their response is not satisfactory at certain points:

  1. Still I do not understant why authors did not follow CLSI quidelines. A major issue in research regarding antimicrobial activity of honey and other honeybee products is that the use of different protocols makes result interpretation rather confusing. MHB perfectly supports MRSA and P. aeruginosa growth. I do not agree that BHI supports better growth.

We have performed new experiments to determine the MIC values of our honey samples, using the broth macrodilution method, following CLSI guidelines. In addition, to make our results comparable with those of other research groups, bacteria were grown in Mueller-Hinton broth instead of BHI. Details of the method were provided in 2.4, while the MIC values obtained with the above method are presented in 3.2 and Table 3 in the revised manuscript.

  1. In my view the implemented  protocol used to assess antibiofilm activity is not well designed. Authors allow bacteria to initiate biofilm formation (4 hours incubation) and then they apply honey to a non-mature biofilm. It is not clear neither if they demonstrate inhibition of biofilm formation nor if they demsontrate eradication of an already formed, mature biofilm.

To create biofilms, we used the protocol of Peeters et al. (2008), who also worked with an incubation time of 4 hours. According to our experience, this time interval was enough for the formation of a significant amount of biofilm, and we mainly concentrated on the inhibition of biofilm formation and not on the eradication of the mature biofilm.

In the future, we would like to work with a longer incubation time, as well, and thereby observe whether honey can significantly reduce the amount of mature biofilm.

Reference:

Peeters E., Nelis  H.J., Coenye T. (2008): Comparison of multiple methods for quantification of microbial biofilms grown in microtiter plates. Journal of Microbiological Methods. 72: 157-165.

  1. Still authors do no provide any data regarding anti-QS activity specifically for MRSA and/or P. aeruginosa. General anti-QS activity of honey using C. violaceum as a model microorganism has been published in several articles (including a very recent one  https://www.mdpi.com/1424-8247/15/10/1212). Unless authors provide novel, specific data on anti-QS activity, I suggest they remove the anti-QS data

We acknowledge that there has been substantial research on the anti-QS activity of various honey types, including acacia, chestnut and linden honeys (Truchado et al. 2009, reference included in our MS), which were also investigated in our study. However, the anti-QS potential of goldenrod and milkweed honeys has not been reported before, which is one of the novelties of our study, as highlighted in lines 446-447 in the revised manuscript.

Since testing the anti-QS activity with C. violaceum as a model organism is widely accepted, we decided to perform this test as the first step in the course of our investigations. After having proved the anti-QS potential of our honey samples in general, in our future research we plan to carry out other assays which will be specific for the bacteria involved in our parallel microbiological experiments.

  1. Authors mention that the novelty of the manucsript has been supplement. They mean Lines 443-448? Elsewhere? Please mention all points throughout manuscript.

In the revised manuscript we highlighted the novelty of our study in lines 401-406 and also in lines 446-447.
